# Evaluation of the Desire to Avoid Pregnancy Scale in the UK: a psychometric analysis including predictive validity

Jennifer Hall [1], Geraldine Barrett [1], Corinne Rocca [2]

¹Reproductive Health Research, University College London, London, UK
²Advancing New Standards in Reproductive Health, University of California, San Francisco, California, USA

**Correspondence to**
Dr Jennifer Hall;
jennifer.hall@ucl.ac.uk

## ABSTRACT

**Objectives** To evaluate the psychometric performance, including predictive validity, of a UK version of the Desire to Avoid Pregnancy (DAP) scale.

**Design** Prospective cohort study for psychometric evaluation.

**Setting** UK.

**Participants** Women in the UK aged 15 years to menopause, who were not pregnant at the time of recruitment in October 2018, were eligible. 994 women completed the baseline survey and 90.2% of women eligible for the 12-month survey participated.

**Primary and secondary outcome measures** The DAP scale was assessed according to key measurement properties of validity (construct (structural and hypothesis testing) and criterion (predictive)), reliability (internal consistency using Cronbach's alpha and test–retest using intraclass correlation coefficients, ICC) and differential item functioning. Item response and classical test theory methods were used.

**Results** The scale was acceptable, understandable and showed good targeting with the full range of scores captured. Construct validity was demonstrated on hypothesis testing, with odds of contraceptive use increasing threefold with each increasing DAP point (range: 0–4). Eighty per cent of women with the lowest DAP score became pregnant within 12 months, compared with <1% of those with the highest DAP score. Reliability, both in terms of internal consistency (Cronbach's α 0.96) and test–retest (ICC 0.95), was excellent. Some tests of structural validity, in relation to model fit with the item-response model, were not met, and investigations suggest further exploration of the factor structure of the DAP is required in other samples. Item 5, regarding relationship with a partner, showed differential item functioning by age, number of children and relationship group.

**Conclusions** The UK DAP is a valid and reliable measure of women's DAP and is highly predictive of pregnancy within the next 12 months. Further evaluations should continue the assessment of the factor structure and the performance of the item relating to the partner.

## INTRODUCTION

Despite increasing recognition of the complexity of the construct of pregnancy preferences and ongoing calls for improved

## STRENGTHS AND LIMITATIONS OF THIS STUDY

⇒ The study was conducted on a large, community-based sample.

⇒ The analysis used both item-response and classical test theory methods.

⇒ This is the first evaluation of the Desire to Avoid Pregnancy (DAP) to look at its predictive validity in terms of pregnancy occurring within 12 months.

⇒ The women in the sample were relatively highly educated, possibly due to recruitment methods using educational establishments and social media.

⇒ We did not ask about gender identity; further work could explore the DAP in relation to gender identity.

measurement,[1–6] there has not, until recently, been a psychometrically validated measure to prospectively assess people's preferences about a potential future pregnancy.[7] This absence has hampered researchers' ability to investigate the underlying causes of unintended pregnancy, including how pregnancy preferences affect contraceptive use, and the consequences of unintended pregnancy for people's health and lives. Practically, it has also meant that there has not been a reliable way to identify who would benefit from contraceptive care and/or preconception advice. This gap has limited our ability to help individuals prevent unintended pregnancies; consequently the proportion of unplanned pregnancies in the UK remains around 45%.[8]

The Desire to Avoid Pregnancy Scale (DAP) was created and evaluated in the USA.[7] Developed using construct modelling and item response theory (IRT) methods, the DAP is a 14-item tool covering three conceptual domains (1): cognitive preferences, (2) affective feelings and (3) practical consequences) of the construct of desire to avoid pregnancy. Each item has a five-point Likert-scale response from strongly agree to strongly disagree, and averaged scores range from 0 (no desire to avoid pregnancy) to 4 (high

desire to avoid pregnancy). Benefits of assessing desire to avoid pregnancy using this measure include that the scale does not assume that people want to or do plan pregnancies, nor that people hold fully formed intentions as it allows the expression of ambiguity and uncertainty.

The rigour of the development and evaluation of the DAP, and the fact that the tool is designed for prospective use (ie, to identify women who do or do not want a pregnancy in the near future) gives hope that the DAP could be a useful tool to support women to avoid the pregnancies that they do not desire and to plan and prepare for those that they do. To use this tool in the UK, it is first necessary to evaluate it in this setting, which is the aim of this study. In addition, we aimed to calculate the predictive validity of the DAP in relation to pregnancy over 1 year.

## METHODS
### Study setting
The Pregnancy Planning Preparation and Prevention (P3 Study) was set up in late-2018 to collect data from non-pregnant women to evaluate the DAP and other questions about pregnancy preferences. Self-reporting females aged 15 and over, living in the UK, who had not gone through the menopause or been sterilised, and who were willing to complete surveys over a 1-year period were eligible to take part. After consenting, women completed a baseline survey online, and they were all invited by email to complete the survey again every 3 months for 1 year unless they had an ongoing pregnancy at two consecutive time points. Data collection ceased in December 2019.

### Survey development
In addition to the DAP scale, the questionnaire included obstetric history, contraception use, reproductive autonomy[9] and sociodemographics. Wherever possible we used existing measures or questions, such as sociodemographic questions recommended by the Office for National Statistics' Harmonisation Strategy[10] and fertility questions from Natsal.[11] The same questionnaire was repeated every 3 months, prefaced with questions asking if the woman was currently pregnant or had been pregnant since the last follow-up.

We conducted cognitive interviews to check women's understanding of the DAP questions and of the whole survey. For this, women were recruited using posters in one central London university (for staff and students), one north London academy (for students aged 15–18 years) and through word of mouth. Women provided written informed consent to take part in the interviews, some of which were face to face (in the university, at the participant's house or in a public meeting space) and some via online video platforms. Interviews were conducted by JH or GB, were audiorecorded and these, plus electronic field notes, were stored in UCL's Data Safe Haven (DSH). The UCL DSH meets relevant information security standards for the secure storage of personal data.

A log was kept of the changes made to the survey; all changes to the DAP were discussed and agreed with CR. We made changes iteratively and retested until we were satisfied that the whole survey was well understood, and no further changes were needed. We conducted 28 cognitive interviews in April-May 2018 with women aged 16–44 from a range of backgrounds and education levels.

The survey was designed and implemented by JH using the REDCap platform,[12 13] hosted on the UCL DSH.

### Sample size
A recent systematic review highlighted the lack of guidance for sample size calculations for psychometric studies.[14] There are two main methods, either a minimum number of subjects or a subject to item ratio, with recommendations from 2 to 20 subjects per item, but with a minimum of 100. The sample does not have to be representative for a valid psychometric analysis.[15] We aimed for 1000 participants (considered excellent on either method) as sufficient to produce stable and accurate model parameters for polytomous IRT models under most data conditions.[16]

We expected that 70% of the cohort would complete follow-up at 12 months[17] and that 10% of them would have a pregnancy in that time[8] giving us 70 pregnancies for our analysis of the predictive validity of the DAP.

A cohort of this size was felt to be feasible to recruit, would meet either criterion for the sample size calculation and would comfortably allow the various tests of reliability and validity, including predictive validity.

### Recruitment
In October 2018, posters and leaflets were distributed in one central London university (for staff and students), one north London academy (for students aged 15–18 years), one outreach sexual health service in south London and one termination service in London. We began recruitment on social media (Facebook and Instagram) through a mixture of sharing through networks and paid for advertising on 15 October 2018. Recruitment was paused on 18 October 2018 and further adverts aimed at women aged under 20 years, non-white, non-Asian, without university-level education and in non-professional occupations went out on 22 October. Recruitment closed on 23 October 2018 when the target of 1000 participants was met. Two weeks later, email invites were sent out to a random subset of women asking them to recomplete the DAP; this survey was set to close when 150 responses had been received. Women received a £5 electronic voucher for every completed survey.

### Data cleaning and validation
A small number of anomalies in the data, for example, unfeasible pregnancy histories or impossible changes in sociodemographic variables, suggested some fabricated data in addition to some genuine errors. These were reviewed on a case-by-case basis, and a set of cleaning rules were agreed by the project steering committee.

Management of records with multiple inconsistencies was agreed by the committee.

## Patient and public involvement

We had public involvement in the development of an overall programme of research on pregnancy planning, now known as the P3 Study, of which this study is one aspect. A patient and public involvement group has recently been established and will be involved in the discussion of how these results are taken forward.

## Analysis

As the DAP was developed using IRT,[7] we sought to conduct the validation using the same methodology. In addition, we carried out some analyses using classical test theory (CTT) and factor analysis, which may be more broadly interpretable than IRT results and may allow future validations to use these methods. We conducted these analyses on the raw average score (range: 0–4) collected at baseline, rather than the model-calculated score, in recognition of how the DAP is likely to be used by most researchers in the field.

### Acceptability, endorsement and targeting

We assessed the acceptability of the DAP through the cognitive interviews and by assessing rates of missing data: >95% completion was considered acceptable. We looked at the overall score distribution characteristics and examined the responses across each item to check none had an endorsement of >80%.

We examined the Wright map (A Wright map, or item-person map, displays the item difficulties and person abilities on the same logit scale and is thus a way of visualising the targeting of the test to the sample and the targeting of individual items to persons) of the DAP scale to see how item and item threshold locations fell relative to participant pregnancy preferences levels. We reviewed the pattern of responses per item to check the targeting of the scale.

### Construct validity (structural validity)

All items were fitted to a partial credit item response model (PCM) and tested for item fit using a weighted mean-squared fit t-statistic of 0.75–1.33 as a guide for good fit.[18] We checked for monotonicity, that is, that for each item, women endorsing higher response categories also had higher scores, on average, on the scale overall. We plotted item characteristic curves to ensure that the responses to each item's categories were ordered.

We also used factor analysis to assess construct validity. Given our findings on PCM item fit, which exhibited some evidence of multidimensionality, we split the data in half randomly and conducted exploratory factor analysis on half the sample, followed by confirmatory factor analysis (CFA) on the other half, as is considered best practice.[19] In both samples, we checked the data were suitable for factor analysis with a polychoric matrix, Bartlett test of sphericity and Kaiser-Meyer-Olkin Measure of Sampling Adequacy.[20] Exploratory item factor analysis using principal factor extraction was conducted on the correlation matrix. Unrotated and oblique rotations were performed, looking for factors with an Eigenvalue greater than one and items with loading of >0.2. CFA with a generalised structural equation model (gSEM) was carried out to compare the one factor model, a two-factor model of positive vs negative items and a three-factor model based on the underlying domains of the DAP scale. As estimates of model fit are not available in STATA for gSEM models, we compared model fit with the Akaike's Information Criteria (AIC) and Bayesian Information Criteria (BIC) where lower AIC/BIC values indicate better fit.[21]

### Construct validity (hypothesis testing)

To test construct validity by hypothesis testing, we hypothesised that women with a higher DAP score would be more likely to use contraception, using baseline data. We created a logistic regression model with self-reported use of any contraception (including all modern contraceptive methods, withdrawal and fertility awareness) in the last 30 days as a binary outcome variable (among women who reported sex with a male partner in the last 30 days) and DAP score as the exposure variable.

### Criterion validity

We assessed the criterion validity of the DAP via its predictive validity with regards to pregnancy occurring within 1 year of completion of the measure at baseline. We generated a binary yes/no variable for whether a woman had had any pregnancies during the 12-month follow-up period, as reported at each of the quarterly surveys, regardless of pregnancy outcome (eg, miscarriage, termination, ectopic, ongoing or other) or timing. This was used as the binary outcome and DAP score as the predictor in a logistic regression model.

### Reliability (internal consistency)

Internal consistency reliability was assessed with the separation reliability coefficient (Person Separation Index) calculated from fitting the item responses to the PCM),[18] the Cronbach's α (>0.7) and by calculating item-rest correlations (to check they were all positive and >0.2).[22 23]

### Reliability (test–retest)

We assessed test–retest reliability with a two-way mixed effects ICC on women who completed a 2-week retest, looking for an ICC >0.8.[24]

### Differential item functioning

We evaluated differential item functioning (DIF) by age (15-24/25-34/35+), ethnicity (white/non-white), relationship group (no relationship/in a relationship not married/married) and number of children (0/1/2/3+) by introducing item-by-trait terms into the IRT measurement model and examining the estimates to see how much the location differed by trait. We used a total logit difference between groups of 0.6 logits as a meaningful difference.[25] Based on findings in the USA, we hypothesised that there would be DIF on item 5 (Becoming

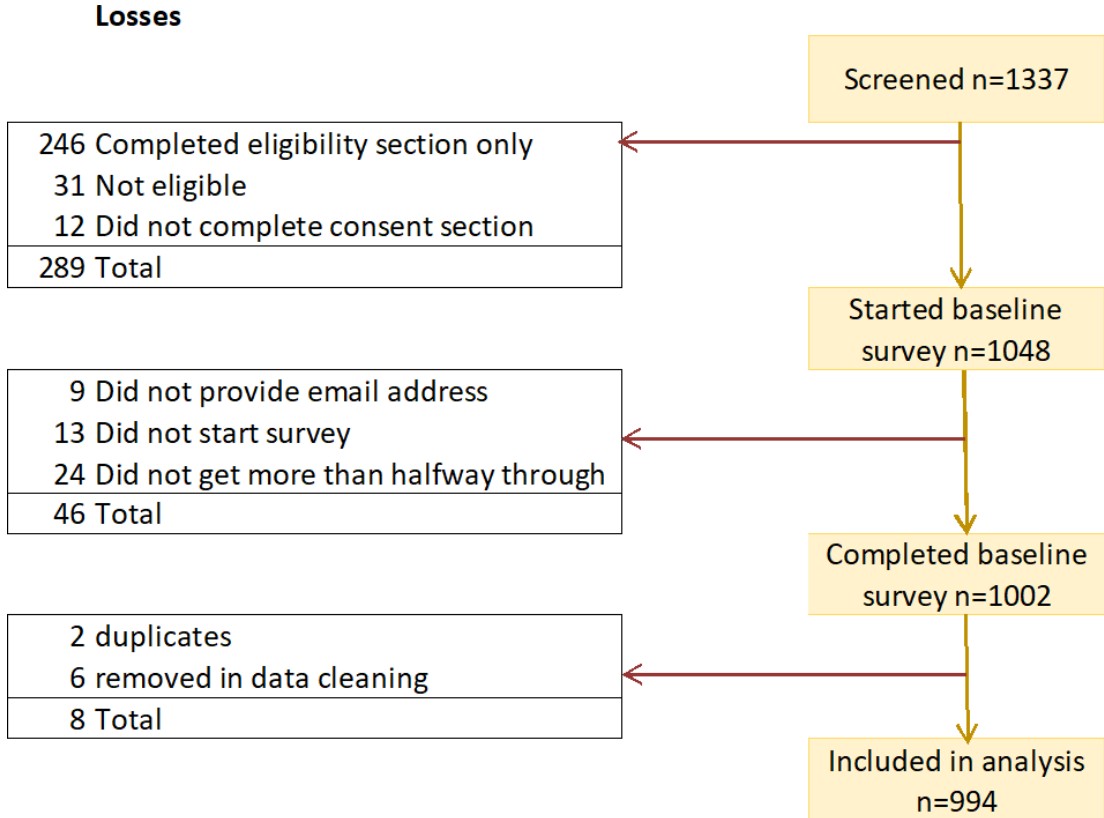

**Losses**

246 Completed eligibility section only
31 Not eligible
12 Did not complete consent section
289 Total

9 Did not provide email address
13 Did not start survey
24 Did not get more than halfway through
46 Total

2 duplicates
6 removed in data cleaning
8 Total

Screened n=1337

Started baseline survey n=1048

Completed baseline survey n=1002

Included in analysis n=994

**Figure 1** Flow chart of recruitment and inclusion of participants.

pregnant in the next 3 months would bring me closer to my main partner) by age, and number of children, but not for ethnicity and for there to be no DIF on any other items.

### Exploratory analysis of factors associated with DAP score

We developed explanatory IRT models for ethnicity, relationship status, age and number of children. We expected no difference in DAP scores for ethnicity and for it to be easier to endorse items (ie, have a higher DAP) if women were not in a relationship, were under 24 or had no children. We checked this by fitting bivariable and multivariable linear regression models.

Analyses were conducted in ACER ConQuest V.5.9.0 and Stata V.17. The data used in this analysis are available in a public repository.[26]

### RESULTS

### Participant characteristics

Figure 1 shows how the cohort of 994 was reached from a total of 1337 records that were created in the online survey software.

Women were aged 15–50 years (median 31, IQR 23–36, mean 29.7), 83.9% were white, 81.6% described themselves as heterosexual and most were married (48.2%) or in a relationship (34.0%) (table 1). A quarter of the women in the cohort had completed school education (ie, school to age 18), 39.1% were educated to undergraduate level and 31.1% had postgraduate or professional

qualifications. More than half (56.7%) had at least one child. The success of the social media recruitment is shown in the distribution of women recruited from across the UK.

The women who completed the 2-week retest (n=152) and those who completed follow-up at 12 months (over 90% of eligible women) were not significantly different by age, relationship status, ethnicity or number of children than those who were not randomly selected into the retest sample than the baseline sample (data not shown).

### Cognitive interviews

The cognitive interviews demonstrated that the DAP was easily understood by women in England. We made minor changes to the syntax that made the flow better when read by women in the UK. The main changes we made were to the wording of the definition of a 'partner'. First, we added a sentence to say 'The next question asks about your partner', tested several iterations of wording for describing what could be considered as a 'partner' and added wording to help women without a current partner to decide how to answer this question. We rephrased 'sexual relations', which women understood to be penetrative sex only, as 'physically intimate'. The wording of the UK DAP is in online supplemental file S1.

### Acceptability, endorsement and targeting

The full range of DAP scores (0–4) was captured; there was a negative skew, as shown in figure 2, with more women having higher levels of DAP. The mean score was

**Table 1** Sociodemographic details of the whole sample at baseline

| | | Freq. | Per cent | | | Freq. | Per cent |
|---|---|---|---|---|---|---|---|
| Age group | 15–19 | 139 | 14 | UK region of residence | Channel Islands | 1 | 0.1 |
| | 20–24 | 143 | 14.4 | | East England | 8 | 0.8 |
| | 25–29 | 139 | 14 | | East Midlands | 80 | 8.05 |
| | 30–34 | 224 | 22.5 | | East of England | 78 | 7.85 |
| | 35–39 | 209 | 21 | | Greater London | 254 | 25.55 |
| | 40–44 | 78 | 7.9 | | North East | 66 | 6.64 |
| | 45+ | 23 | 2.3 | | North West | 81 | 8.15 |
| | Missing | 39 | 3.9 | | Northern Ireland | 16 | 1.61 |
| Relationship | Not in a relationship | 152 | 15.3 | | Scotland | 83 | 8.35 |
| | In a relationship, but not married | 338 | 34.0 | | South East | 123 | 12.37 |
| | Married | 479 | 48.2 | | South West | 94 | 9.46 |
| | Other | 1 | 0.1 | | Wales | 35 | 3.52 |
| | Missing | 24 | 2.4 | | West Midlands | 60 | 6.04 |
| Race/ethnicity | White | 834 | 83.9 | | Missing | 15 | 1.51 |
| | Mixed | 26 | 2.6 | No of children | 0 | 430 | 43.3 |
| | Asian | 69 | 6.9 | | 1 | 208 | 20.9 |
| | Black | 24 | 2.4 | | 2 | 233 | 23.4 |
| | Other | 9 | 0.9 | | 3+ | 78 | 7.9 |
| | Prefer not to say/ Missing | 32 | 3.2 | | Missing | 45 | 4.5 |
| Completed level of education | School | 247 | 24.9 | Sexuality | Heterosexual | 811 | 81.6 |
| | Undergraduate | 389 | 39.1 | | Not heterosexual | 146 | 14.7 |
| | Postgraduate | 309 | 31.1 | | Prefer not to say/ missing | 37 | 3.7 |
| | Other | 18 | 1.8 | | | | |
| | Missing | 31 | 3.1 | | | | |

2.53, SD 1.05, median 2.71 and IQR 1.86–3.36. Although all questions were optional in our online survey, there was over 95% completion for every DAP item and very little missing data. No item category had >80% endorsement (online supplemental file S2).

The Wright map (online supplemental file S3) showed the full range of participant locations with item

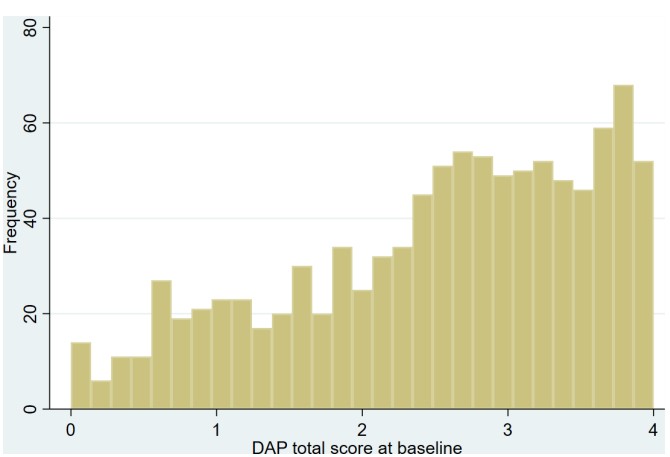

**Figure 2** Frequency distribution of DAP score at baseline. DAP, Desire to Avoid Pregnancy.

thresholds falling at all levels of participants' pregnancy preferences, demonstrating good targeting and that the items captured the full range of the construct.

### Construct validity (structural validity)
#### Item fit to model
Most items fit the PCM, based on the weighted mean square statistic being between 0.76 and 1.33, but five items did not (online supplemental file S4). It was noted that the two items with misfit indicating greater deviance than expected due to higher discrimination than the model predicted (items with a weighted mean fit of <0.75) were both positively worded items in domain one, cognitive preferences. Conversely, the three items with poorer fit due to lower discrimination (items with a weighted mean fit of >1.33) were in domain three; anticipated practical consequences, and two of them were negatively worded. Item 5, about the partner, had the poorest fit to model, consistent with the original development.[7] Repeating the item-fit analysis with either a two factor (positive vs negative items) or three factor (domains) model reduced the item misfit.

Monotonicity was confirmed, that is that average DAP scores increased among those endorsing each increasing

response category on each individual item, indicating excellent internal structure. Item characteristic curves confirmed that the responses to each item's categories were ordered.

### Factor analysis

Tests confirmed the split datasets (n=497 in each) were suitable for factor analysis (Bartlett test of sphericity (p<0.001) and Kaiser-Meyer-Olkin Measure of Sampling Adequacy (0.960)). EFA using principal factor extraction with no rotation conducted on the polychoric factor matrix suggested a one factor model with an Eigenvalue of 9.84 (LR test <0.001) and where this factor explained 91% of the variability in responses. All items strongly loaded on to this factor (loadings >0.7 for all except item 5 which was 0.51). Oblique rotation did not improve model fit.

CFA was conducted the one factor model proposed by the EFA and the two and three factor models suggested by the IRT analysis. There was very little difference in the AIC or BIC between the models; the two-factor (positive v negative) was the best fit and three-factor model (domains) was almost identical, followed by the one factor model, however, the differences were small.

### Construct validity (hypothesis testing)

As per our hypothesis, among women who had had sex with a male partner in the last 30 days, those with higher DAP scores (greater desire to avoid pregnancy) were significantly more likely to report contraception use in the last 30 days (OR 3.20 95% CI 2.58 to 3.98). A predicted 29.5% of women with a DAP score of 0 were using contraception (95% CI 20.6% to 38.5%), compared with 98% (95% CI 96.6% to 98.9%) for women with a DAP score of 4.

### Criterion validity

Over 90% of women eligible for follow-up at 12 months completed the survey and 139 women experienced at least one pregnancy during the year of the cohort. For every one-point increase in DAP score there were 0.22 the odds

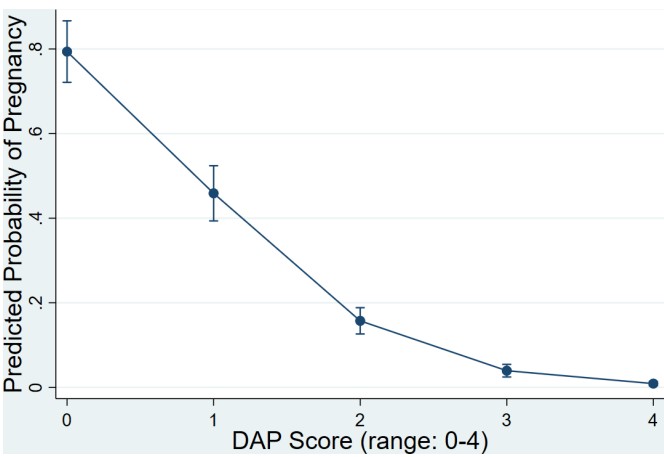

**Figure 3** Probability of pregnancy within 1 year as predicted by DAP score. DAP, Desire to Avoid Pregnancy.

of becoming pregnant within 12 months (95% CI 0.17 to 0.28). As shown in figure 3, women with a DAP score of zero had an 80% (95% CI 72.1% to 86.7%) probability of becoming pregnant within a year, compared with 0.89% (95% CI 0.36% to 1.44%) with a score of 4.

### Reliability (internal consistency)

The Person Separation Reliability coefficient was 0.91 and Cronbach's α was 0.96, both extremely high. All inter-item correlations were positive and all item-total correlations were positive and greater than 0.2. Item 5, 'Becoming pregnant in the next 3 months would bring me closer to my (main) partner' had the lowest item-test and item-rest correlations (0.506 and 0.441 respectively) and the scale's internal consistency was not reduced (ie, α remained 0.96) when excluding it.

### Reliability (test-retest)

The ICC was 0.95 (95% CI 0.92 to 0.97) indicating excellent stability over the 2 weeks test–retest period.

### Differential item functioning

As expected, we found evidence of DIF on item 5 (Becoming pregnant in the next 3 months would bring me closer to my main partner) by age, number of children, and relationship group. For example, compared with participants aged 25–34 years with otherwise similar DAP scores, those aged 15–24 were more likely to agree to the statement. There was also evidence of DIF on item 13 (If I had a baby in the next year, it would be hard for me to manage raising the child) by age, number of children and relationship group, with younger women more likely to agree to the statement than older women with otherwise similar DAP scores and those with more children or in a relationship less likely to agree with the statement than those with fewer children or not in a relationship.

### Exploratory analysis of factors associated with DAP score

Both the eIRM and linear regressions showed that there was no significant difference in overall DAP score by ethnicity (β 0.18, 95% CI –0.02 to 0.38) but there were differences by age, relationship status, number of children and education level. A multivariable linear regression model was created; ethnicity remained non-significant and was excluded from the model; all other variables remained statistically significant, as shown in table 2.

### DISCUSSION

The DAP scale generally performed well in the UK population. Cognitive interviews and the high completion rates showed good understanding and acceptability to women. The DAP items captured the full range of participants' pregnancy preference levels, and the scale showed good targeting and acceptable levels of endorsement across item response options. In terms of construct (structural) validity, mean scores increased by each item's increasing categories and the item categories' peaks were ordered on the item characteristic curves. Our construct

**Table 2** Multivariable linear regression of characteristics associated with DAP score

| Variable | | Coefficient | 95% CI | P value |
|---|---|---|---|---|
| Relationship status | Baseline—married | | | <0.001 |
| | In a relationship, not married | 0.48 | 0.30 to 0.65 | |
| | Not in a relationship | 0.77 | 0.53 to 1.00 | |
| Age | Baseline 15–24 | | | <0.001 |
| | 25 | −0.63 | −0.84 to −0.42 | |
| | 35 | −0.50 | −0.75 to −0.26 | |
| No of children | Baseline—0 | | | <0.001 |
| | 1 | −0.22 | −0.42 to −0.01 | |
| | 2 | 0.63 | 0.40 to 0.85 | |
| | 3 | 0.81 | 0.53 to 1.09 | |
| Level of education | Baseline—school | | | 0.043 |
| | Undergraduate | −0.17 | −0.35 to 0.00 | |
| | Postgraduate | −0.04 | −0.23 to 0.16 | |
| | Other | 0.32 | −0.13 to 0.77 | |

DAP, Desire to Avoid Pregnancy.

validity hypothesis test confirmed that for every one-point increase in DAP score, women were three times as likely to be using contraception. In addition, the DAP score was highly predictive of pregnancy, with 80% of those with a DAP score of zero likely to become pregnant within a year, compared with <1% of those with a DAP score of four. Reliability, both in terms of internal consistency and test–retest, was excellent.

However, there were a few areas where our expectations were not met. Based on the item-fit estimates to the PCM, there was some suggestion that the DAP scale may not be unidimensional. While EFA suggested a single factor model, CFA found that a two-factor model (ie, treating positively and negatively worded items as separate factors) fitted marginally better than a three-factor model (ie, treating each of the DAP's conceptual domains as separate factors), which fitted marginally better than one factor. From a psychometric perspective this suggests that there may be different dimensions within the DAP scale, rather than it being unidimensional. Nevertheless, the differences in AIC/BIC were very small and we are limited by the lack of other goodness of fit statistics available in STATA, so we cannot confidently recommend an improved structure. Notably, when fitting the DAP items to a multidimensional PCM treating positively and negatively worded items as different dimensions, all items other than item 5 met criteria for item fit. The DAP is a new measure; this is the first evaluation outside the USA to be published though we are aware of other studies underway. We suggest that these studies conduct further analysis of the factor structure and dimensionality of the DAP based on our findings. Regardless the DAP appears to be functioning well as a measure, predicting pregnancy and passing the standard CTT-based tests.

There were similarities between the UK and USA in that the item that was hardest to endorse was the same in both settings (Item 9: 'It would be the end of the world for me to have a baby in the next year').[7] In the UK sample, however, women found it easier to agree with the items in the conceptual domain of 'anticipated practical consequences' (items 5, 12, 13 and 14) meaning they felt that having a baby in the next year would have a greater impact on their lives, in terms of practical consequences, than the women in the US sample. For example, on item 12 (I would feel a loss of freedom if I had a baby in the next year) 35% of women in the US sample agreed or strongly agreed, compared with 64% in the UK sample. Comparing these items between the two samples suggests a real difference in how different groups of women think about the effects that childbearing would have on their lives under their current circumstances. This may be explained in part by important differences between the populations studied in the UK and USA. The US sample was recruited from reproductive and primary health facilities and targeted women 'at risk' of pregnancy. Women in the US study were younger (41%<25 compared with 28%), less likely to be white (16% v 84%), married (26% vs 48%) or to have completed college-level education (10% vs 70%) and were slightly more likely to already have a child (61% vs 57%).[7] These are all factors which are known to influence pregnancy preferences and translated to a different distribution of the DAP score, with women in the UK sample expressing a higher desire to avoid pregnancy (averaged raw scores: USA mean 2.2, SD 1.1 vs UK mean 2.5, SD 1.1).

Our analyses pointed to relatively poor psychometric performance of item 5 (pregnancy would bring me closer to my main partner) in several ways, consistent with the

USA study.[7] In our cognitive interviews, respondents without a partner found the item difficult to respond to; some opted for the neutral 'neither agree nor disagree' response while others selected 'strongly disagree' as they had no partner to become closer to. Item 5 was the item we made the most changes to in terms of the wording, much of which was aimed at how women without a partner should think about this question. Psychometrically, the item did not fit the PCM, and illustrated notable DIF by several sociodemographic variables. Although partnership factors are a key component shaping an individual's pregnancy preferences, it is likely that people's perceptions of how a pregnancy would affect a relationship could be reflected in their responses to other DAP items. Removal of this item should be considered and, as our analysis showed, would not reduce the high internal consistency of the scale.

Multivariable analysis has shown that women's desire to avoid pregnancy is associated with age, relationship status and number of children, which is consistent with the wider literature on the determinants of pregnancy intention.[8 27–32] Future analyses could consider how desire to avoid pregnancy changes over time and as external factors, such as relationship status and education level, change.

## Strengths and limitations

This is a large study with data collected from women across the UK in the community, rather than from healthcare settings. We captured the full spectrum of desire to avoid pregnancy and are the first to assess the properties of the DAP scale using IRT and CTT methods outside the USA. We conducted a test–retest analysis and, uniquely, examined the predictive validity of the DAP by looking at pregnancy over the subsequent year.

Our recruitment strategy may have contributed to the sample being more educated than average, as we used a college and university as recruitment sites. Furthermore, social media recruitment can result in an over-representation of young white women and disadvantages those without internet access.[33 34] While the proportion of women from a black or ethnic minority background was commensurate with the general population, in absolute terms this was only 160 women, just 15 of whom experienced a pregnancy during this study. Further work should be conducted in non-white women. In this study participants self-reported that they were female, and we did not ask about gender identity; it will be important to understand how the DAP works for people of all genders.

## Conclusion

The UK DAP is a valid and reliable measure of desire to avoid pregnancy and a useful tool for research. CTT tests showed that the measure was highly valid and reliable based on established standards. While not all preset criteria were met for the IRT tests, the criteria were met when considering the possibility of multidimensionality. Future evaluations of the DAP should consider removal of item 5 and examine different factor structures in light of our findings. We are the first to show that the DAP score is highly predictive of pregnancy. This is of particular use to those attempting to recruit a preconception cohort or, conversely, women who are unlikely to become pregnant, for example, for a pharmacological study. The DAP may also have a potential clinical use, to guide preconception or contraception care. In order to be practical, consideration should be given to a shortened version to make it more amenable for use in busy, time-constrained clinical settings.

**Acknowledgements** We would like to thank Dr Mark Wilson for his support around item response theory-based analyses and Anasztazia Gubijev who supported the follow up of the cohort. We would also like to thank the project steering committee (Professor Judith Stephenson, Professor Anna David, Dr Sue Mann, Dr Sarah Crozier, Dr Natalie Edelman, Dr Anita Coutinho and James Harris) for their input on the data cleaning decisions and overall guidance.

**Contributors** JH: guarantor, conceptualisation, funding acquisition, methodology, investigation, data curation, formal analysis, writing-original draft, writing-review and editing, project administration. GB: conceptualisation, methodology, investigation, writing-review and editing. CR: conceptualisation, methodology, formal analysis, writing-review and editing.

**Funding** The study was funded by an NIHR Advanced Fellowship held by JH (PDF-2017-10-021).

**Competing interests** None declared.

**Patient and public involvement** Patients and/or the public were involved in the design, or conduct, or reporting, or dissemination plans of this research. Refer to the Methods section for further details.

**Patient consent for publication** Not applicable.

**Ethics approval** We received ethical approval from the UCL Research Ethics Committee for both the development and testing of the survey (ref 3974.002) and for the online survey with 12-month follow-up (ref 3974.003).

**Provenance and peer review** Not commissioned; externally peer reviewed.

**Data availability statement** Data are available in a public, open access repository. The dataset is available in the UCL Research Data Repository DOI: 10.5522/04/20195081.

**ORCID iDs**
Jennifer Hall http://orcid.org/0000-0002-2084-9568
Geraldine Barrett http://orcid.org/0000-0002-9738-1051
Corinne Rocca http://orcid.org/0000-0001-9892-249X

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
