## [Reviewer comments · BMJ Open]

ARTICLE DETAILS

TITLE (PROVISIONAL)	Evaluation of the Desire to Avoid Pregnancy Scale in the United Kingdom: a psychometric analysis including predictive validity
AUTHORS	Hall, Jennifer; Barrett, Geraldine; Rocca, Corinne

VERSION 1 – REVIEW

REVIEWER	Heikinheimo, Oskari University of Helsinki, Obstetrics and gynecology
REVIEW RETURNED	23-Jan-2022

GENERAL COMMENTS	This is an interesting study focusing on the use and validation of a USA derived questionnaire on pregnancy intendedness in the UK. The number of study subjects is large, and appears sufficient for the purpose of the study. There is plenty of statistical modeling, which may be of limited interest to the general readership. Thus, speciality journal (such as BMJ journal of sexual & reproductive health) might be another option. Alternatively, some of the statistical modeling could be presented as supplementary material. My questions concern the study flow, registration the pregnancies during the study and use of contraception. Please provide a study flow as a figure. Did all women complete the survey at 3mo intervals (page 6, line 8)? If so, which score was used for the study (the 1st one)? Moreover, was there much intraindividual variation in the DAP score over the year? Which factors were associated with possible changes in DAP score? Changes in relationship status? Were there any exclusion criteria (such as history of surgical sterilization or hysterectomy)? How were the pregnancies monitored / recorded? Self-reporting by the study subjects or use of health care registries? Were all pregnancies - those resulting in delivery, spontaneous abortion (ectopics) or induced abortion included? Was there variation between the different types of pregnancy outcomes (e.g. need of induced abortion vs delivery) according to DAP scores? How was the use of contraception defined? Were all methods (such as fertility awareness, barrier vs short acting hormonal (such as pill or ring) vs long acting reversible methods (such as IUD or implant)) similarly included? Of the group having DAP score of 0 80% became pregnant. This high rate is to be expected among fertile aged women having regular unprotected heterosexual intercourse, yet some 30% of them reported use of contraception
--

	(most likely not very effective contraception...). Conversely, less than 1% of the women having score of 4 had any pregnancies during the 1y of follow-up. Assuming that they were sexually active, they must have used very effective contraception (IUDs, implants). Thus would be interesting to know how the use of different methods varied according to the DAP scores. Developing a shorter, more practical version of the questionnaire to be used by public and to guide clinical work is applauded.
--	---

REVIEWER	Aksu, Sıdıka Gazi University Faculty of Health Sciences, Department of Nursing
REVIEW RETURNED	03-Jun-2022

GENERAL COMMENTS	It's a nice, field-contributing scale, and I found it very nice and interesting to follow the scale result and investigate its consistency. I think your article can be published and will contribute to the field, but I have a few suggestions.  1. Your references are very old, I want them updated. 2. Are the groups homogeneous in the demographic characteristics table in the analysis? I want to see it and I want it to be mentioned in the table. 3. Can the tables be differentiated for easy reading? 4. Did you use a checklist for reporting the article? I didn't see this? I want you to specify if it has been used. Even if it is not used, please revise and upload the article using the checklist appropriate for the research type.
--

VERSION 1 – AUTHOR RESPONSE

Reviewer: 1
Dr. Oskari Heikinheimo, University of Helsinki

Comments to the Author:

This is an interesting study focusing on the use and validation of a USA derived questionnaire on pregnancy intendedness in the UK. The number of study subjects is large, and appears sufficient for the purpose of the study. Thank you

There is plenty of statistical modeling, which may be of limited interest to the general readership. Thus, speciality journal (such as BMJ journal of sexual & reproductive health) might be another option. Alternatively, some of the statistical modeling could be presented as supplementary material. As a validation paper, the statistics, while extensive, are crucial and central to the paper, hence we have not put them in a supplementary file. However, in response to this comment, we have made changes throughout the Analyses section to simplify language. We did consider BMJ SRH, but they do not have a strong track record of publishing validation studies and are not fully open access.

My questions concern the study flow, registration the pregnancies during the study and use of contraception.

Please provide a study flow as a figure. Did all women complete the survey at 3mo intervals (page 6, line 8)? If so, which score was used for the study (the 1st one)? We had not included a study flow chart as the analysis focuses almost exclusively on the baseline DAP data. We have made this

clearer in the paper (line 157-8) and have added more detail on follow up at 12m, which is only used for predictive validity, in lines 240-3 and also in lines 92-4.

Moreover, was there much intraindividual variation in the DAP score over the year? Which factors were associated with possible changes in DAP score? Changes in relationship status? These are all pertinent and interesting questions and we have analyses underway looking at these very things. However, before we can explore if, how and why the DAP score changes over time we must first validate the instrument for use in this population so that we are sure that it is a psychometrically robust measure. We have stated in the paper that these are important future analyses (lines 385-6)

Were there any exclusion criteria (such as history of surgical sterilization or hysterectomy)? As mentioned (line 91), women who were postmenopausal were excluded. Women who had been sterilized were also excluded and we have added this to the text line 91.

How were the pregnancies monitored / recorded? Self-reporting by the study subjects or use of health care registries? Were all pregnancies - those resulting in delivery, spontaneous abortion (ectopics) or induced abortion included? Pregnancies were self-reported by women at each three-month follow up and all pregnancies were included regardless of outcome. We have added more detail in lines 194-199.

Was there variation between the different types of pregnancy outcomes (e.g. need of induced abortion vs delivery) according to DAP scores? We did not follow up women to delivery so we do not have data on this. For the purposes of the validation study it was sufficient to know whether they became pregnant or not.

How was the use of contraception defined? Were all methods (such as fertility awareness, barrier vs short acting hormonal (such as pill or ring) vs long acting reversible methods (such as IUD or implant)) similarly included? Contraception use was self-defined and all methods were included in a multi-choice question for women to select from. We have added more detail to the manuscript in line 190-2.

Of the group having DAP score of 0 80% became pregnant. This high rate is to be expected among fertile aged women having regular unprotected heterosexual intercourse, yet some 30% of them reported use of contraception (most likely not very effective contraception...). Conversely, less than 1% of the women having score of 4 had any pregnancies during the 1y of follow-up. Assuming that they were sexually active, they must have used very effective contraception (IUDs, implants). Thus would be interesting to know how the use of different methods varied according to the DAP scores. You are absolutely right, and again this is a planned analysis! Previous work with the DAP in the USA has shown that it is associated with use and consistency of use, but not necessarily type (<https://www.ncbi.nlm.nih.gov/pmc/articles/PMC7028518/>).

Developing a shorter, more practical version of the questionnaire to be used by public and to guide clinical work is applauded.

Reviewer: 2

Dr. Sıdıka Aksu, Gazi University Faculty of Health Sciences

Comments to the Author:

It's a nice, field-contributing scale, and I found it very nice and interesting to follow the scale result and investigate its consistency. I think your article can be published and will contribute to the field, but I have a few suggestions. Thank you

1. Your references are very old, I want them updated. We think that here you must be referring to some of the references about the criteria in the methods section? These are the standard references for the field and as such are the appropriate ones to reference as they are still correct, regardless of their age. Our methods references do include those as recent as 2019.

2. 2. Are the groups homogeneous in the demographic characteristics table in the analysis? I want to see it and I want it to be mentioned in the table.

We are not sure which groups/ table this comment is referring to. Table 1 is the demographic characteristics of the whole sample, so there are no groups. We have clarified in the caption that this is for the whole sample at baseline. There are no groups in the analysis, other than where we randomly split the data in half for EFA and CFA, and we have already included confirmation that the two-week follow up and 12m follow up groups were not different on key socio-demographics or DAP score. If we have misinterpreted this comment, please let us know where we should direct our attention.

3. Can the tables be differentiated for easy reading? We have tried to improve the look of the tables and we will happily reformat them in line with any BMJ Open guidance, but could not find any online.

4. Did you use a checklist for reporting the article? I didn't see this? I want you to specify if it has been used. Even if it is not used, please revise and upload the article using the checklist appropriate for the research type.

In the absence of the availability of a self-complete checklist document, we have added a separate file indicating how we have met the COSMIN reporting guidelines for studies on measurement properties of patient reported outcome measures. Our responses should be read in conjunction with the pdf of the checklist which we have uploaded and is available here: https://www.cosmin.nl/wp-content/uploads/COSMIN-reporting-guideline_1.pdf

VERSION 2 – REVIEW

REVIEWER	Heikinheimo, Oskari University of Helsinki, Obstetrics and gynecology
REVIEW RETURNED	06-Jul-2022

GENERAL COMMENTS	The paper has been nicely revised. Three minor suggestions:
---

	 - Ending the Abstract with 'item 5' is a bit blunt. (many people read just the abstract). Consider adding the meaning of item 5 - Full meaning of the term 'gender' might not be obvious to all, at least non-English speakers. Consider using 'perceived gender'. - Supplementary file 1. Consider adding item numbers (Item 1 etc) before/after each item to facilitate reading of the table.
--	---

VERSION 2 – AUTHOR RESPONSE

Reviewer: 1

Dr. Oskari Heikinheimo, University of Helsinki Comments to the Author:

The paper has been nicely revised. **Thank you**

Three minor suggestions:

- Ending the Abstract with 'item 5' is a bit blunt. (many people read just the abstract). Consider adding the meaning of item 5 **We have amended this to read: Further evaluations should continue the assessment of the factor structure and the performance of the item relating to partner.**

- Full meaning of the term 'gender' might not be obvious to all, at least non-English speakers. Consider using 'perceived gender'. **We are aware that this is a contentious topic at the moment, and that there is not yet a universally accepted form of words. We have discussed this and have chosen not to use 'perceived gender' and instead have used 'gender identity', which we hope will be acceptable. The sentence in question now reads: 'We did not ask about gender identity; further work could explore the DAP in relation to gender identity.' (lines 56-7) and at the end of the discussion: 'In this study participants self-reported that they were female, and we did not ask about gender identity; it will be important to understand how the DAP works for people of all genders'. (lines 402-3)**

- Supplementary file 1. Consider adding item numbers (Item 1 etc) before/after each item to facilitate reading of the table. **We have added the item numbers as requested.**